# The Relationship between Mindfulness and Social Adaptation among Migrant Children in China: The Sequential Mediating Effect of Self-Esteem and Resilience

**DOI:** 10.3390/ijerph192316241

**Published:** 2022-12-04

**Authors:** Yue Wang, Zexin Zheng, Xiaoyi Duan, Mengsha Li, Ying Li

**Affiliations:** School of Education, Zhengzhou University, Zhengzhou 450001, China

**Keywords:** migrant children, mindfulness, self-esteem, resilience, social adaptation

## Abstract

Social adaptation of migrant children is not only related to the physical and mental health and development of individuals, but also reflects the level of urban social integration and stable development. Mindfulness has a protective effect on individual social adaptation. Self-esteem and resilience were found to be positively associated with mindfulness and social adaptation. Based on the Positive Youth Development Perspective, this study aimed to explore whether self-esteem and resilience sequentially mediated the associations among mindfulness and social adaptation. A total of 526 migrant children were assessed with the questionnaires regarding mindfulness, self-esteem, resilience, and social adaptation. The results indicated that mindfulness was positively associated with social adaptation of migrant children. Self-esteem and resilience played the sequential mediating roles between mindfulness and social adaptation. The present study revealed the influence and mechanism of mindfulness on social adaptation and provided some guidance for the intervention programs to promote migrant children’s adaptability.

## 1. Introduction

With the rapid advancement of urbanization and the expanding of floating population in China, the characteristics of migration are changing from “individual migration” to “family migration”, and from “temporary residence” to “long-term residence” [1], resulting in a large number of migrant children. Migrant children refer to school-age children and adolescents who registered their household registration (Hukou system) in other provinces (autonomous regions or cities) or rural areas of local province, and migrate with their parents and live with them in local cities, and receive education at local schools [2]. According to the statistics of the China Ministry of Education in 2018, the population of migrant children in compulsory education had reached 19.52 million in China, of which 14.24 million were children of migrant workers who moved to cities with their families [3]. As a result of migration, migrant families leave behind important social ties with relatives, close friends, and neighborhoods. Such losses of social contact can exert considerable influence on migrant children who experience crucial developmental periods in their current lifespan [4]. Furthermore, due to China’s household registration system, the migrant population does not have equitable opportunities with local residents in public schools, health care, housing, and other social welfare [5]. They are also vulnerable to exclusion by consumption, social relations, culture, and the welfare system [6]. Therefore, migrant children are likely to face greater challenges both externally and internally, and are prone to development and adaptation problems, compared with local children. 

Social adaptation refers to the process and state where individuals achieve a balance between the social environment and themselves through self-regulation and self-management and coping with the external environment in the process of interaction with the social environment [7]. A previous study confirmed that migrant children might be confronted by more social adaptation problems, which lead to more internal and external problems. For example, compared with urban children, previous studies have shown that migrant children show higher levels of loneliness, social anxiety and depression, lower life satisfaction and subjective well-being, as well as more problem behaviors and threat perception [8]. Following those studies, migrant children’s social adaptation problems should not be underestimated [9].

The social adaptation of migrant children reflects the level of social integration and stable development [10]. It is of far-reaching practical importance to explore the underlying antecedents of social adaptation and how they influence children’s social adaptation. Previous studies have shown that mindfulness could positively predict individual adaptability [11]. Furthermore, the correlation between mindfulness and socio-cultural adaptation is consistent in cross-cultural studies [12]. Therefore, this study will explore the influence of mindfulness on migrant children’s social adaptation and its internal mechanism, which can not only theoretically broaden the scope of research on the influencing factors of migrant children’s social adaptation, but also provides a new direction for the intervention of migrant children’s social adaptation.

### 1.1. Mindfulness and Social Adaptation of Migrant Children

Previous research on migrant children was mainly based on the perspective of the deficit model [13]. The Positive Youth Development Perspective, which focuses on the advantages and potential of adolescents, provides a novel idea for the migrant children’s social adaptation [14]. This perspective emphasizes that more attention should be paid to the positive psychological characteristics, psychological states, and mechanisms of migrant children. Among so many factors, mindfulness has received widespread focus for its health-promoting and adaptive effects.

Mindfulness refers to an individual’s non-judgmental attention and awareness of current events and experiences, and emphasizes the importance of being open and receptive to present experiences without judgment or habitual reactive patterns [15]. This acceptance is not passive submission to the current environment, but a positive response to the present experience [16]. The Mindfulness and Awareness Model [15] suggests that individuals can process psychological content in de-automated and separate ways through mindfulness, which can help individuals develop the ability to gain insight into their current thoughts and emotional responses, enhancing their flexibility among cognition-emotion-behavior. In this way, individuals can experience internal and external stimuli more objectively and accept internal and external experiences more openly. Previous studies have shown that it is more efficient and easier for individuals with high levels of mindfulness to regulate emotions and behaviors and form friendships in a new cultural environment [17], so as to adapt to the new living environment better. In addition, in the group of migrant children, higher levels of mindfulness were associated with reduced loneliness, anxiety, depression, and improved life satisfaction and subjective well-being [18,19]. However, few studies investigated the positive effect of mindfulness on social adaptation in a group of migrant children.

On account of the above theories and existing empirical studies, mindfulness may act as a buffer for migrant children to make psychological adjustments and develop healthy social functioning. Therefore, we hypothesize that mindfulness can positively predict the social adaptation of migrant children (Hypothesis 1).

### 1.2. The Mediating Role of Self-Esteem

How does mindfulness relate to individuals’ social adaption? Self-esteem may be an important mediating variable. Self-esteem refers to an individual’s positive or negative attitude towards himself or herself, reflecting the individual’s degree of self-acceptance and self-respect [20]. As one of the core components of an individual’s self-system, self-esteem is directly related to an individual’s psychological well-being, with a wide range of effects on an individual’s cognition, motivation, emotion, and social behavior [21,22]. The Social Bonding Theory states that a high level of self-esteem plays a protective role when individuals are exposed to risk factors [23,24] and is an important factor in reducing social maladjustment [25]. People with high self-esteem are more likely to avoid the influence of anxiety, which means they can keep internal consistency and stability when challenged by a new environment. Previous literature has also suggested that adolescents with high self-esteem tend to have better social adjustment skills [26]. It follows that self-esteem plays an important role in an individual’s adaptation to a new social environment, and is also a meaningful indicator of social adaptation status.

On the other hand, previous studies also suggested that mindfulness is closely linked to self-esteem. As a positive trait, mindfulness emphasizes openness and acceptance, not only in the way we treat others and external things, but also in the way we treat ourselves [27]. It is more likely for individuals with higher levels of mindfulness to accept and cope with their internal and external environments in positive ways [28]. Several relevant empirical studies have shown that mindfulness can positively influence individual development through the increase of individual self-esteem levels. According to Bajaj and colleagues [29], individuals with high levels of mindfulness may have higher self-esteem, which helps to reduce anxiety and depression. Moreover, a study by Jia and colleagues [19] also revealed that mindfulness can significantly improve individuals’ self-esteem levels. Based on the above analysis, this study suggests that self-esteem may play a mediating role between mindfulness and the social adaptation of migrant children. In other words, mindfulness can indirectly influence social adaptation of migrant children through self-esteem (Hypothesis 2).

### 1.3. The Mediating Role of Resilience

Resilience seems to be an important factor for migrant children to overcome adaptation difficulties and achieve good social adaptation. Resilience refers to the ability or trait of an individual to effectively adapt to life adversity, such as pressure, setbacks, and trauma [30]. The Quality–Pressure Model indicated that individuals with different characteristics show differences in adaptability [31]. Individuals with high levels of resilience have lower vulnerability qualities and more psychological resources, so they are less likely to be negatively affected by stress and more adaptable in adverse situations [32,33]. Therefore, in migrant children, resilience might be helpful in building their social adaptation abilities.

In addition, the association between mindfulness and resilience has been confirmed by many studies. Zenner (2014) has shown that mindfulness training can improve individuals’ cognitive functioning and resilience levels [34]. Zhou et al. (2017) also found a positive correlation between mindfulness and resilience in Chinese children from grade 3 to grade 6 [35]. It is noteworthy that a study found that mindfulness exerts a positive impact on psychological components (including subjective well-being and life satisfaction), which is highly correlated with adaptive ability through the mediation effect of resilience [36]. Thus, we hypothesized that resilience would play a mediating role between mindfulness and social adaptation, which is to say that mindfulness can indirectly influence the social adaptation of migrant children through resilience (Hypothesis 3).

### 1.4. Sequential Mediating Effects of Self-Esteem and Resilience

In addition to the independent mediating role of self-esteem and psychological resilience, the relationship between self-esteem and resilience is also closely associated in migrant children’s social adaptation. The resilience protection model theory claims that protective factors promote the development of individual resilience through buffering or mitigating the negative effects of risk factors on individual development [37]. Wright and Masten (2005) identified that children’s self-assurance (self-confidence, high self-esteem, and self-efficacy) serves as protective factors in mitigating risk [38]. The stronger these protective factors are, the easier it will be for the individual to mitigate the influences of negative experiences and enhance their resilience. Conversely, if the protective factors of the individual are not sufficient to counter the risk factors, the resilience of the individual will be damaged [39]. As for migrant children, a study proved that self-esteem can effectively alleviate or offset the negative impact of adverse circumstances, and improve the level of resilience [40].

Overall, individuals with a higher level of mindfulness hold more positive attitudes towards their own abilities and values [41], they have stronger self-regulation and emotional regulation ability to cope with adversity and trauma, and predict a higher resilience together with a reduced possibility of maladaptation [42,43]. However, to date, few studies have investigated the role of both self-esteem and resilience as sequential mediators of the relationship of mindfulness with social adaption in migrant children. Thus, we expected that self-esteem and resilience would play a sequential mediating role in the relationship between mindfulness and social adaptation of migrant children (Hypothesis 4), and the hypothetical model was shown in Figure 1.

## 2. Materials and Methods

### 2.1. Participants and Procedure

In this study, we recruited students from two elementary schools and two junior middle schools in a developed city in Henan province located in the central part of China. Henan is a large agricultural province and is one of the most populous provinces in China. A large number of rural people flocked to provincial capitals in search of career advancement, resulting in numerous migrant children. After obtaining informed consent from children, their guardians, and teachers, a total of 1485 students from grade 5 to grade 8 finished the demographic questionnaires. From the 1242 (83.6%) provided valid questionnaires, we eventually selected 526 participants who met the standards of migrant children (according to the definition of migrant children, the standard is that they are not registered in Zhengzhou, but they have lived in Zhengzhou for more than half a year with their parents or other guardians and are in the stage of compulsory education). After that, the interviewer used the uniform questionnaire and instructions to conduct the test. All the questionnaires were filled out anonymously and collected on the spot after their completion.

### 2.2. Measures

Mindfulness was measured with the Mindfulness Attention Awareness Scale (MAAS), which was translated and revised by Chen and colleagues [44]. There are 15 items in the scale, such as “I find it difficult to focus on what is happening at the moment”. The items were rated on a 6-point Likert scale, ranging from 1 = “almost never” to 6 = “almost always”. The scale examines an individual’s level of mindfulness, which involves cognitive, emotional, and physiological aspects in daily life, and the higher the score, the higher the level of mindfulness. In the present study, Cronbach’s α of the scale was 0.77.

Self-esteem was measured with the Self-Esteem Scale (SES), which was translated and revised by Ji and Yu [45]. There are 10 items in the scale, such as “I feel that I am a person of value”. The items were rated on a 4-point Likert scale, ranging from 1 = “strongly disagree” to 4 = “strongly agree”, and item 8 is a reverse score. The scale assesses overall feelings of self-worth and self-acceptance: a higher score represents a higher self-esteem of the individual. In the present study, Cronbach’s α of the scale was 0.86.

Resilience was measured with The Resilience Scale for Chinese Adolescents (RSCA), which was translated and revised by Hu and Gan [46]. There are 27 items in the scale, such as “I tend to be more mature and experienced after setbacks”. The scale is divided into two dimensions, manpower and support force, which can be subdivided into 5 factors. The sub-scales of goal focus, emotion control, and positive cognition belong to manpower, and family support and interpersonal assistance sub-scales belong to support force. The items were rated on a 5-point Likert scale, ranging from 1 = “strongly disagree” to 5 = “strongly agree”, and the higher the score, the higher the level of resilience. Holistic Cronbach’s α of the scale was 0.91, while the Cronbach’s α of goal focus, emotion control, positive cognition, family support, and interpersonal assistance ranged from 0.65 to 0.86.

Social adaptation was measured with the Social Adaptation Scale for Children and Adolescents (SASCA), which was created by Hu [47]. SASCA was specifically created for Chinese migrant children and teenagers in order to satisfy their education needs. There are 48 items in the scale, such as “I know how to make more friends”. The scale was composed of eight first-order factors (learning autonomy, environment satisfaction, activity participation, life independence, interpersonal coordination, interpersonal friendliness, social identity, and social vitality) and three second-order factors (social relations and concept adaptation, learning and school adaptation, and life and activity adaptation). The items were rated on a 5-point Likert scale, ranging from 1 = “strongly disagree” to 5 = “strongly agree”, and the higher the score, the higher the social adaptation of the individual. Holistic Cronbach’s α of the scale was 0.91, and the Cronbach’s α of learning autonomy, environment satisfaction, activity participation, life independence, interpersonal coordination, interpersonal friendliness, social identity, and social vitality ranged from 0.64 to 0.88.

### 2.3. Data Analysis

SPSS 21.0 was used for data collation and analysis in two steps. Firstly, data were analyzed using descriptive statistics and zero-order correlation (Pearson correlation) analysis. Secondly, Hayes’ (2013) PROCESS for SPSS macro (Model 6) was used to establish a multiple mediation model of self-esteem and resilience in the association between mindfulness and social adaptation among migrant children [48], which means all possible paths from authentic self-presentation to depression in Figure 1 were examined by Model 6. The analysis of Model 6 in PROCESS revealed direct effects, indirect effects, and bootstrap (repeated sampling 1000 times) confidence intervals (CIs) for indirect effects. If CI did not include zero, then the effect would be significant [49].

### 2.4. Check for Common Method Bias

Since self-reported data collection methods may cause common method bias problems, this study adopted methods emphasizing anonymous survey and reverse scoring for some items to carry out some procedural controls. Meanwhile, Harman’s single factor test was adopted for all project principal component factor analyses of the questionnaire. The results show that in 100 projects, the characteristic root which is greater than 1 public factor is 26, and the highest factor of explanation variance is 19.45%, less than 40% of the critical standards, which proves that there is no obvious problem of common method biases in this study [50].

## 3. Results

### Descriptive Statistics and Correlations

A total of 526 students participated in this study. The mean age of participants was 12.06 (SD = 1.18), and ranged from 10 to 15. These respondents consisted of 271 boys (51.5%) and 255 girls (48.5%); 132 (25.1%) were from grade 5, 117 (22.2%) were from grade 6, 142 (27.0%) were from grade 7, and 135 (25.7%) were from grade 8 (Table 1). The results of participants’ scores of psychological measurement and Pearson correlation analysis showed that there was a significant positive correlation between mindfulness, self-esteem, resilience, and social adaptation (*p* < 0.001), and the correlation strength between variables of interest is neither too weak nor too strong, which means we could further perform regression analysis. See Table 2 for details.

The regression analysis results (Table 3) showed that the effect of mindfulness on social adaptation (*β* = 0.15, *p* < 0.001), self-esteem (*β* = 0.32, *p* < 0.001), and resilience (*β* = 0.34, *p* < 0.001) had a significant positive predictive effect. The correlation between self-esteem and resilience (*β* = 0.57, *p* < 0.001) and social adaptation (*β* = 0.33, *p* < 0.001) had a significant positive predictive effect. Resilience versus social adaptation (*β* = 0.26, *p* < 0.001) had a significant positive predictive effect.

Bootstrap results (Table 4) indicated that self-esteem and resilience played mediating roles between mindfulness and migrant children’s social adaptation, with a mediating effect value of 0.24, accounting for 60% of the total effect of mindfulness on social adaptation (0.40).

The mediating effect is composed of the indirect effect of the following three paths: the first path is mindfulness → self-esteem → social adaptation, the effect value is 0.11, and the indirect effect accounts for 27.50% of the total effect. The confidence interval of the 95% interval of Bootstrap is [0.07, 0.15], excluding 0 value, indicating that the first path has a significant mediating effect. The second path is mindfulness → self-esteem → resilience → social adaptation, the effect value is 0.05, and the indirect effect accounts for 12.50% of the total effect. The confidence interval of the 95% interval of Bootstrap is [0.03, 0.08], which does not include 0 value, indicating that the second path has a significant mediating effect. The third path is mindfulness → resilience → social adaptation, the effect value is 0.09, and the indirect effect accounts for 22.50% of the total effect. The confidence interval of the 95% interval of Bootstrap is [0.06, 0.13], which does not include 0 value, indicating that the third path has a significant mediating effect.

In conclusion, self-esteem and resilience play a serial mediating role between mindfulness and social adaptation, that is, mindfulness can not only directly affect migrant children’s social adaptation, but also indirectly influence their social adaptation through self-esteem and resilience (see Table 3 and Figure 2 for details).

## 4. Discussion

The social adaptation of migrant children is not only related to the well-being of the migrant population, but also crucial to social harmony and stability as well as economic development. Previous studies have found that high levels of mindfulness, good self-esteem, resilience, and other factors contribute to the good adaptation of individuals, and there is a strong link between mindfulness, self-esteem, and resilience. To our knowledge, this is the first study to examine the role of self-esteem and resilience as sequential mediators of mindfulness and social adaption in Chinese migrant children. Therefore, combined with the protective effects of self-esteem and resilience, this study investigated the internal influence mechanism of mindfulness on the social adaptation of migrant children. Based on the sample of Chinese migrant children, this study helps us to gain an understanding of the underlying mechanism between mindfulness and social adaptation, which informs us of possible interventions to promote social adaption in migrant children.

### 4.1. The Impact of Mindfulness on Social Adaptation

As hypothesized, this study indicated that there is an association of mindfulness with social adaption in migrant children, which is similar to the result of a previous study [11]. The mindfulness re-perception model indicates that mindfulness can improve the attention function of children effectively, alter their perception of undesirable stimuli, and also increase tolerance and acceptance of internal or external environments [51]. Our result indicated that similar relationships between mindfulness and acceptance of circumstance could exist in migrant children.

According to Bronfenbrenner’s ecological systems theory [52], a series of interacting environmental systems that are immediately and closely connected to an individual are crucial for the individual’s physical well-being and mental growth. Accompanied by the involvement of subjective factors, the systems interact with individuals and influence individual development. Individuals who have experienced or are experiencing severe stress/adversity consciously or unconsciously assess their own abilities, situation, and interpersonal relationships, while individuals with high levels of self-awareness and agency have positive psychological adaptation outcomes [53]. Our result verified that such interactions between subjective awareness (e.g., mindfulness) and psychological adaptation outcomes (social adaptation) could also be applied to migrant children who are facing dramatic changes in their living and learning environments. Children of a high mindfulness level could more effectively evaluate their own situation, use internal resources and external support to promote their own development and positive psychological adaptation, and are likely to have better social adaptation.

### 4.2. The Mediating Effect of Self-Esteem and Resilience

Consistent with Hypothesis 2, self-esteem plays a mediating role between the mindfulness and social adaptation of migrant children. The acceptance and openness involved in mindfulness can reduce depressive thoughts caused by negative events, and to a certain extent, avoid negative automatic thinking patterns towards oneself [54]. Therefore, individuals, including migrant children, with high levels of mindfulness are more likely to be predictive of high levels of self-esteem in adverse environments, and high self-esteem will in turn play a protective role in the face of risk factors for them.

The functions of self-esteem include self-protection, experience interpretation, determination of individual expectations, and even influencing the development of a healthy personality [55]. Compared with those who have lower self-esteem, adolescents with higher self-esteem are more likely to be good at controlling and regulating their emotions and behaviors, which then promotes the development of social adjustment [56]. Individuals with different levels of self-esteem may give different interpretations of the same experiential situation, while our results proved that migrant children with high self-esteem are more likely to be associated with more positive attitudes towards their abilities and values, and tend to be predictive of a positive attitude towards life, which is represented by higher social adaptation abilities.

The results of the study also verified Hypothesis 3, that mindfulness could indirectly influence the social adaptation of migrant children via resilience. In a general way, high mindfulness predicts good resilience, which corresponds to previous research findings [57]. Individual cognition makes a difference in the developmental mechanisms of resilience, and mindfulness can improve attention and cognitive functioning through nonjudgmental awareness. Meanwhile, mindfulness will lead to a positive bias that tends to focus on positive stimuli and having a more positive cognitive evaluation of neutral stimuli [58], and such positive interpretation bias could positively predict resilience [59].

Apart from this, high resilience predicts good social adaptation, which is consistent with the results of previous studies as well as the model of Quality–Pressure [60]. Resilience can assist an individual in coping with stressful situations and reflects their adaptability to adversity [43], with the level of mental resilience determining an individual’s ability to cope and adapt to stressful situations [61]. Therefore, our study indicated that migrant children with high resilience are able to draw on psychological resources in a timely manner, adopt positive and effective coping strategies, and enhance their ability to deal with the stress of urban life [62] and eventually show a positive association with good social adaptation.

It is important to acknowledge that the relation of mindfulness to social adaption was sequentially mediated by self-esteem and resilience. This result suggests that self-esteem and resilience are closely related, supporting previous findings [39] and confirming the resilience protection model theory [37]. Compared with those who have lower self-esteem, migrant children with higher self-esteem tend to be predictive of a high level of self-acceptance, which means they are more confident and have a greater sense of self-efficacy. Those are the guarantees of good resilience. In other words, it is likely that self-esteem, as a protective effect of resilience, can help individuals recover quickly from dilemma and thus mitigate the adverse effects, with such association also being evident in migrant children based on the present study findings. In conclusion, for migrant children, high self-esteem as a protective factor of resilience might effectively buffer or offset the negative effects of disadvantages and is positively associated with their psychological resilience level.

### 4.3. Implication and Limitation

Based on the above findings, the present study provides significant implications for the intervention of social adaption in migrant children. Findings from studies like this may be interesting to parents, educators, and government that wish to develop intervention strategies of social adaption for migrant children under disadvantaged situations.

The results of the study reveal that we should not ignore the plasticity and developmental possibilities of migrant children in the education process, and should value migrant children’s own positive emotional experiences and personality traits for their social adaptation, discover and cultivate their inner positive emotional experiences and personality qualities, and help migrant children use their inner psychological resources to achieve good social adaptation. Further, school mental health workers can integrate mindfulness into the school psychology curriculum by conducting group counseling and training activities that include mindfulness content and methods to improve the social adaptation of migrant children by intervening in their level of mindfulness.

A few study limitations should, however, be noted. First, this is a cross-sectional study and causality cannot therefore be determined. Additionally, due to the nature of a cross-sectional design, data on variables of interest were collected during the same time period, which may have introduced some unwanted noise to our model. The current design nevertheless provides some insights into the investigated research question and informative directions for future studies. Specifically, further investigations can aim to examine the proposed model with a longitudinal design or adopt intervention experiments. Second, this study focused on a specific sample, migrant children in China, due to research interest. The generalizability of our findings is therefore limited. Whether children and adolescents generally showed similar results remains to be determined by future studies. Separately, we did not conduct a control group design or longitudinal design, so our conclusion cannot infer the difference between migrant children and local children or the effectiveness of the intervention proposed. Future studies should pay more attention to that. Finally, as pointed out earlier, there may be other mediators in the relationship between mindfulness and social adaption that we did not consider. This study only focuses on the individual characteristics of migrant children and has little understanding of their family and social environment. Future studies should extend our findings by measuring other theoretically relevant factors.

## 5. Conclusions

In conclusion, the current study has contributed to a key point of understanding of the sequential mediation model between mindfulness and social adaptation in a sample of migrant children. We found that resilience and self-esteem serve as a potential mechanism in the relationship between mindfulness and social adaptation. In particular, resilience and self-esteem mediate the link between mindfulness and social adaptation among Chinese migrant children not only in parallel but also sequentially.

## Figures and Tables

**Figure 1 ijerph-19-16241-f001:**
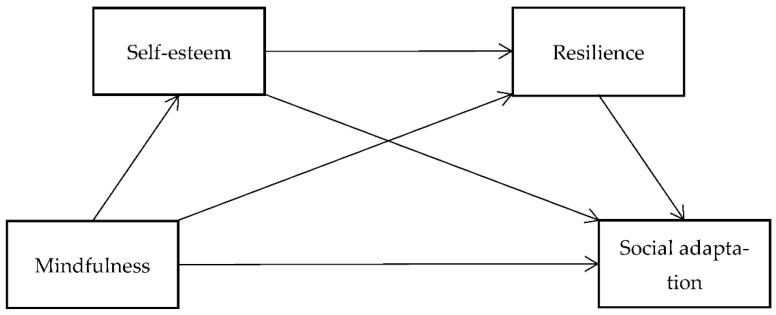
The hypothetical model of the impact of mindfulness on social adaptation.

**Figure 2 ijerph-19-16241-f002:**
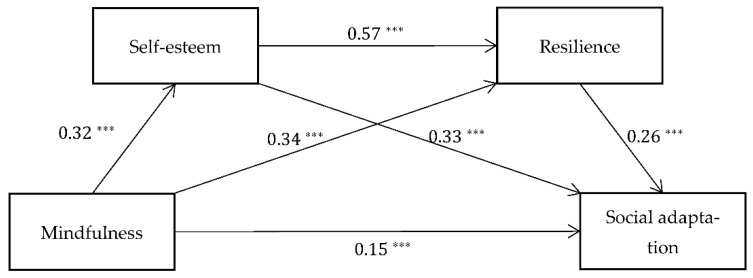
The sequential mediation model between mindfulness and social adaptation through self-esteem and resilience. *** *p* < 0.001.

**Table 1 ijerph-19-16241-t001:** Participants’ characteristics (n = 526).

	Age	Gender		Grade			
		Male	Female	Grade 5	Grade 6	Grade 7	Grade 8
**Mean or n** **± SD or %**	**12.06** **1.18**	271	255	132	117	142	135
		51.5%	48.5%	25.1%	22.2%	27.0%	25.7%

**Table 2 ijerph-19-16241-t002:** Summary of means, standard deviations, and correlations of variables of interest.

	M (Overall)	SD (Overall)	1	2	3	4
1. Mindfulness	4.60 (68.96)	0.58 (8.67)	1			
2. Self-esteem	3.07 (30.68)	0.47 (16.97)	0.40 ***	1		
3. Resilience	3.59 (97.04)	0.63 (4.71)	0.49 ***	0.55 ***	1	
4. Social Adaptation	3.70 (147.95)	0.53 (21.14)	0.45 ***	0.54 ***	0.57 ***	1

*** *p* < 0.001.

**Table 3 ijerph-19-16241-t003:** Standardized results of the moderated mediation model.

Outcome	Predictor	Overall Fit Index			Significance of Regression Coefficients		
		*R*	*R* ^2^	*F*	*β*	*SE*	*t*
self-esteem resilience	mindfulness	0.43	0.18	29.50 ***	0.32	0.03	9.61 ***
	mindfulness	0.62	0.39	66.54 ***	0.34	0.04	8.22 ***
	self-esteem				0.57	0.05	11.19 ***
social adaptation	mindfulness	0.67	0.45	71.47 ***	0.15	0.04	4.29 ***
	self-esteem				0.33	0.04	7.41 ***
	resilience				0.26	0.03	7.48 ***

*** *p* < 0.001.

**Table 4 ijerph-19-16241-t004:** Bootstrap results of mediating effect.

Pathways	Point Estimates	Bootstrap *SE*	Bootstrap 95% CL		Relative Mediating Effect Proportion
			BootLLCI	BootULCI	
Total indirect effect	0.24	0.03	0.19	0.30	60.00%
Mindfulness → self-esteem → social adaptation	0.11	0.02	0.07	0.15	27.50%
Mindfulness → self-esteem → resilience → social adaptation	0.05	0.01	0.03	0.08	12.50%
Mindfulness → resilience → social adaptation	0.09	0.02	0.06	0.13	22.50%

## Data Availability

The original contributions presented in the study are included in the article; further inquiries can be directed to the corresponding author/s.

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
