# Peer review of "The Relationship between Mindfulness and Social Adaptation among Migrant Children in China: The Sequential Mediating Effect of Self-Esteem and Resilience"

_ijerph, 2022, doi:10.3390/ijerph192316241_

Round 1

Reviewer 1 Report (Previous Reviewer 2)

This manuscript examined the relationship between self-efficacy and resilience to social adaption and mindfulness in children of immigrants in China. When developing an intervention program for children's mental health, this study will be a reference for deciding what to make content.

#1 Introduction.

Please fix the structure of the introduction section to be concise and readable. It's hard to read because the paragraphs have different topics. For example, the second paragraph uses previous research on the psychological characteristics of immigrant children, followed by a definition of social adjustment. In 1.3., the authors wrote two separate sentences about people with high levels of resilience, but it seems to be able to combine.

About the authors' hypothesis

Self-efficacy is a model that predicts resilience, but doesn't resilience predict self-efficacy?

#2. Methods

Let me ask about ethical considerations. This study was intended for minors. The authors mentioned that IC was conducted, but did it also conduct for guardians?

About statistical analysis. Please define the strength of the correlation coefficient. If the absolute value of the correlation coefficient is between 0.0 and 0.2, the interpretation is that there is little correlation, and so on.

#3. Results

In the outcome row of Table 3, does the resilience position meet here?

#4. Discussion

Please write a discussion considering the definition of the strength of the correlation coefficient.

Author Response

This manuscript examined the relationship between self-efficacy and resilience to social adaption and mindfulness in children of immigrants in China. When developing an intervention program for children's mental health, this study will be a reference for deciding what to make content.

#1 Introduction.

Please fix the structure of the introduction section to be concise and readable. It's hard to read because the paragraphs have different topics. For example, the second paragraph uses previous research on the psychological characteristics of immigrant children, followed by a definition of social adjustment. In 1.3., the authors wrote two separate sentences about people with high levels of resilience, but it seems to be able to combine.

Reply: The construct of our article have been revised. Firstly, we improved the readability via making one paragraph corresponded to one topic. Then, we refine and streamline our introduction of variables which could improve the concision of the paper.During this process, we also streamlined the references to ensure that each reference is highly relevant to the content.

About the authors' hypothesis

Self-efficacy is a model that predicts resilience, but doesn't resilience predict self-efficacy?

Reply: Firstly, the mediating variables examined in our study is “self-esteem” and “resilience”, so we didn’t clarify the relationships between “self-efficacy” and “resilience”. Besides, you might be confused by the direction of the relationships within self-esteem and resilience. We put forward this hypothesis that self-esteem predict resilience is because some theory model (e.g. resilience protection model theory) and studies have indicated that protective factors (self-esteem) could promote the resilience. However, we didn’t reject the probability that resilience might influence self-esteem, given that the regression analysis could not precisely illustrate the casual effect between variables.

#2. Methods

Let me ask about ethical considerations. This study was intended for minors. The authors mentioned that IC was conducted, but did it also conduct for guardians?

Reply: The IC was conducted for children and their guardians, and we also asked for their teacher to evaluate the suitability of questionnaires. Such contents have been added in newly revised manuscripts.

About statistical analysis. Please define the strength of the correlation coefficient. If the absolute value of the correlation coefficient is between 0.0 and 0.2, the interpretation is that there is little correlation, and so on.

Reply: the purpose of correlation analysis in our manuscripts is to describe the general characteristics and relationships between the variables, which are prerequisites for our further regression analysis. The correlations strength between our variables are neither little (as you mentioned, between 0.0 and 0.2), nor too strong (r > 0.7) which could lead to Multicollinearity problems (See graph below). Therefore we didn’t define the strength and their meanings of correlation coefficient in detail. According to your comments, we added the supplementary sentence in manuscripts in order to specify the function of correlation analysis.

M(overall)

SD(overall)

1

2

3

4

5

6

1.Mindfulness

4.60(68.96)

0.58(8.67)

1

2.Self-esteem

3.07(30.68)

0.47(16.97)

0.40***

1

3.Resilience

3.59(97.04)

0.63(4.71)

0.49***

0.55***

1

4.Social Adaptation

3.70(147.95)

0.53(21.14)

0.45***

0.54***

0.57***

1

#3. Results

In the outcome row of Table 3, does the resilience position meet here?

Reply: The position of resilience is according to the table template of previous sequential mediation studies. We also refer to a study published in IJERPH: “Relationship between Intergenerational Emotional Support and  Subjective Well-Being among Elderly Migrants in China: The Mediating Role of Loneliness and Self-Esteem” https://www.mdpi.com/1660-4601/19/21/14567

#4. Discussion

Please write a discussion considering the definition of the strength of the correlation coefficient.

Reply: As mentioned in method part, the purpose of correlation analysis in our manuscripts is to make show that the relationships between variables are qualified to further regression analysis. Given our regression analysis have described the association and mechanism between variables in detail, we don’t think adding the discussion of correlation strength in discussion part is constructive to our manuscripts but only to replicate the discussed results.

Reviewer 2 Report (Previous Reviewer 1)

Dear authors.

I thank the authors for the opportunity to review the manuscript again.

You have placed the additions I recommended about age characteristics in "Implication and Limitation". It would be better if you discussed the age aspect in a general text." Discussion". This is acceptable.

But in any case, there are no more comments from my side.

Author Response

Thank you for your valuable comments, which will be very important for us to further improve the manuscript.

Round 2

Reviewer 1 Report (Previous Reviewer 2)

Thank you for revising the manuscript. The revised manuscript and your comments make it easier to understand.

This manuscript is a resubmission of an earlier submission. The following is a list of the peer review reports and author responses from that submission.

Round 1

Reviewer 1 Report

The work is of interest. However, there are a number of remarks to the research plan.

1) There is no comparison group consisting of their children - non-migrants who were born in Zhengzhou, permanently reside, registered there. Therefore, we cannot draw conclusions specifically about migrant children, perhaps these are general patterns of childhood.

2) Specify in the methods which correlation analysis was used.

3) To assess the indirectness of influence, it is better to use factorial analysis of variance ANOVA. Regression and correlation analysis reveals it insufficiently.

Author Response

Thanks for your comments. After considering your suggestions carefully, we have revised our manuscripts point by point. Responses toward your comments will be given in following contents, and the detailed revision could be checked in updated manuscripts.

Reviewer 1                                                                       

#1: There is no comparison group consisting of their children-non-migrants who were born in Zhengzhou, permanently reside, registered there. Therefore, we cannot draw conclusions specifically about migrant children, perhaps these are general patterns of childhood.

Response: We have revised statements which draw conclusion specifically about migrant children. Although our conclusion couldn’t specify the problems which migrant children are faced with without comparison with common children (e.g. local), the underlying relationships between mindfulness and social adaptation could be still inspiring when dealing with the problems of social adaptation. Our study mainly focus on the underlying relationships between these variables based on the sample of migrant children in China. Given previous study indicating the poor social adaptation of migrants children (In introduction section), our results could be applied to mitigate such difficulties. However, your comments directly point out a limitation of our study, so we have added that in our limitation.

#2: Specify in the methods which correlation analysis was used.

Response: The method of correlation (Pearson Correlation) is clarified in our updated manuscript.

#3: To assess the indirectness of influence, it is better to use factorial analysis of variance ANOVA. Regression and correlation analysis reveals it insufficiently.

Response: For ANOVA analysis, the present study found difficulties in designing and applying suitable paradigm which can exclude unrelated factors in order to investigate the influence function between mindfulness and social adaptation in migrants children, because it’s time-costly and unpractical for short term study. Therefore, our study focus on the predictive relationships between mindfulness and social adaptation among migrants children. Unfortunately, we didn’t pay enough attention to our statement which leads to some misunderstandings of our study’s purpose. Thanks for your comments and we have revised our statements in updated manuscript.

Reviewer 2 Report

This manuscript is a study of the relationship between mindfulness and social adjustment in Chinese migrant children, investigating the mediation of self-esteem and resilience.

China is a vast country, and even if people migrate within the country, it can be imagined that the culture may be very different depending on the destination, which may present similar mental health issues as returnee children. Examining factors related to adjustment will help in the development of future clinical intervention tools. However, the manuscript needs to be reviewed in several respects.

#1:Please specify the research design. If it is an observational study, there is a STROBE statement. It is easier for most readers to follow the guidelines appropriate to the study design.

#2:For the methods section, from line 201 to 211, it states that 1242 children were asked valid questions and 526 children were selected. In the next paragraph, the authors wrote: "after obtaining informed consent". Who were the subjects of this study? Was it the aforementioned 526 children? Of the 526 children, were only those who gave consent included?

#3:In the Results section. Only the results of the analysis on correlations were listed, but please write down the characters of the research subjects and the scores of the psychological measurements (MAAS, Resilience, SASCA), using tables as well.

#4:Both the Introduction section and the Discussion section, it seems to duplicate the same content in some of the sentences. It lengthens the entire manuscript and makes it difficult to understand. Also, psychological assessment scores were unknown, that makes it difficult to interpret the results.

Author Response

Thanks for your comments. After considering your suggestions carefully, we have revised our manuscripts point by point. Responses toward your comments will be given in following contents, and the detailed revision could be checked in updated manuscripts.

Reviewer 2                                                                                      

#1: Please specify the research design. If it is an observational study, there is a STROBE statement. It is easier for most readers to follow the guidelines appropriate to the study design.

Response: We feel sorry that we didn’t specify our research design clearly in previous manuscripts. Given that an observational study is difficult for us to exclude irrelevant factors, we apply a cross-sectional study based on questionnaires which is highly validated and qualified in previous studies. Following your comments, we have provided some statements which illustrated our design in detail in our updated manuscript.

#2: For the methods section, from line 201 to 211, it states that 1242 children were asked valid questions and 526 children were selected. In the next paragraph, the authors wrote: "after obtaining informed consent". Who were the subjects of this study? Was it the aforementioned 526 children? Of the 526 children, were only those who gave consent included?

Response: We appreciate that you pointed out those mistakes which could be deleterious to the credibility of our research’s procedure, and we have corrected our statements in updated manuscript. Only 526 participants whose background were qualified for migrant children definition were included in only analysis, while we had investigated 1242 children. Of course, all 1242 children were informed with consent, but we didn’t clearly state that in our previous manuscripts which leads to some misunderstanding. We felt sorry for that.

#3: In the Results section. Only the results of the analysis on correlations were listed, but please write down the characters of the research subjects and the scores of the psychological measurements (MAAS, Resilience, SASCA), using tables as well.

Response: We have provided the details of research subjects’ characters and the scores of the psychological measurements in newly updated manuscripts.

#4: Both the Introduction section and the Discussion section, it seems to duplicate the same content in some of the sentences. It lengthens the entire manuscript and makes it difficult to understand. Also, psychological assessment scores were unknown, that makes it difficult to interpret the results.

Response: In updated manuscript, we have pruned our introduction and discussion section and make sure that the readers won’t feel burdensome and confused when reading them. We also provided the psychological measurements scores in recent uploaded manuscript as mention above (#Response 3), and we hope that could be helpful for readers to understand the results. Unfortunately, we couldn’t provide any professional psychological assessment (e.g. mental disorders assessment) because such questionnaires should be conducted and interpreted by professional psychological doctors (we are not specialized in such field). Besides that, we also tended to explain our results in a general aspect rather than in a specific group by controlling some factors as covariates. However, before fulfilling questionnaires, we inquired the children’s teachers about their cognitive function and understanding ability and made sure those children fully understand the items of questionnaires after tests.

Round 2

Reviewer 1 Report

Dear authors. Thanks for the replies. I accept your explanation. They clarify a lot. However, the absence of a comparison group is a major drawback. Because it is impossible to draw conclusions about a group without comparing it. Restrictions are not enough. I understand that the study has already been done and if the original design of the study did not include a comparison group, then there is nowhere for it to come from.

I suggest doing so. Find an article. which studies the adaptation of ordinary (local) children. I think such studies exist. In discussing the results of your article, give a theoretical comparison of your results with literature data. This will partially solve the issue.

Reviewer 2 Report

Thank you for your response to the peer review. However, the manuscript does not appear to have been adequately revised.

#1. The authors answered that added the study design, but there is no description in the methods section. The STROBE statement is guidance that is also used for a cross-sectional study. It should be better to attach a checklist of STROBE statements as an appendix.

 #2. Regarding psychological rating scale scores, Table 2 appears to list each question's item mean and SD, not the scale's overall score. As there are no mean and SD for each scale, it is not possible to determine the validity of the content of the discussion.